# Viromes of Tabanids from Russia

**DOI:** 10.3390/v15122368

**Published:** 2023-11-30

**Authors:** Alexander G. Litov, Oxana A. Belova, Ivan S. Kholodilov, Anna S. Kalyanova, Magomed N. Gadzhikurbanov, Anastasia A. Rogova, Larissa V. Gmyl, Galina G. Karganova

**Affiliations:** 1Laboratory of Biology of Arboviruses, FSASI Chumakov Federal Scientific Center for Research and Development of Immune-and-Biological Products of RAS, 108819 Moscow, Russia; novosti-wxo@yandex.ru (A.G.L.); mikasusha@bk.ru (O.A.B.); ivan-kholodilov@bk.ru (I.S.K.); magomed_19@mail.ru (M.N.G.); rogova_aa@chumakovs.su (A.A.R.); lvgmyl@mail.ru (L.V.G.); 2Department of Biology, Lomonosov Moscow State University, 119991 Moscow, Russia; 3Institute for Translational Medicine and Biotechnology, Sechenov University, 119991 Moscow, Russia

**Keywords:** Tabanidae, *Narnaviridae*, *Totiviridae*, *Flaviviridae*, *Xinmoviridae*, *Permutotetraviridae*, negevirus, *Rhabdoviridae*, *Solemoviridae*, *Picornavirales*

## Abstract

Advances in sequencing technologies and bioinformatics have greatly enhanced our knowledge of virus biodiversity. Currently, the viromes of hematophagous invertebrates, such as mosquitoes and ixodid ticks, are being actively studied. Tabanidae (Diptera) are a widespread family, with members mostly known for their persistent hematophagous behavior. They transmit viral, bacterial, and other pathogens, both biologically and mechanically. However, tabanid viromes remain severely understudied. In this study, we used high-throughput sequencing to describe the viromes of several species in the *Hybomitra*, *Tabanus*, *Chrysops*, and *Haematopota* genera, which were collected in two distant parts of Russia: the Primorye Territory and Ryazan Region. We assembled fourteen full coding genomes of novel viruses, four partial coding genomes, as well as several fragmented viral sequences, which presumably belong to another twelve new viruses. All the discovered viruses were tested for their ability to replicate in mammalian porcine embryo kidney (PEK), tick HAE/CTVM8, and mosquito C6/36 cell lines. In total, 16 viruses were detected in at least one cell culture after three passages (for PEK and C6/36) or 3 weeks of persistence in HAE/CTVM8. However, in the majority of cases, qPCR showed a decline in virus load over time.

## 1. Introduction

Tabanidae (Diptera) are a cosmopolitan family, with members mostly being nuisance pests for people and livestock because of their painful bites and persistent biting behavior [1]. The Tabanidae family is more diverse than any other hematophagous insect family and includes more than 4000 described species [2,3]. In Russia, 114 species have been described, with six genera being the most represented: *Tabanus*, *Atylotus*, *Heptatoma*, *Chrysops*, *Haematopota,* and *Hybomitra* [4].

Adult tabanids are fast fliers and can cover a distance of up to 2 km daily. Both males and females use sugars of plant origin, such as nectar, to provide energy for flight. Most females seek a blood meal after mating in order to produce eggs, with the size of blood meals varying from 20 μL for small species up to 600 μL for larger species [1].

Pastured cattle, wildlife species, and even humans suffer from tabanid attacks. In addition to blood loss from feeding, tabanids cause extreme annoyance. Large numbers of tabanids in an area can reduce weight gain and milk production in cattle. For instance, in French Guiana, the mean daily weight gain for cattle during the season of horsefly activity is 418 g, which is 327 g less than the annual average [5]. In Russia, authors report losses of up to 30% in milk production and reductions of as much as 45% in weight gained during tabanid season [4]. Tabanids can cause additional economic losses due to their impact on human outdoor recreational activities, such as trekking, fishing, swimming, and camping [1].

Tabanid biology and feeding behavior make them suitable vectors for the transmission of viral, bacterial, and other pathogens [1,5]. The transmission of the filarial nematodes *Loa loa* [6], *Elaeophora schneideri* [7], *Dirofilaria roemeri* [8], and *Dirofilaria repens*, as well as the protozoa *Haemoproteus metchnikovi* and *Trypanosoma theileri*, involves disease agent replication or development within tabanids [1]. Mechanical transmission by tabanids (primarily *Chrysops* spp., *Hybomitra* spp., and *Tabanus* spp.) plays a major role in the transmission of the equine infectious anemia virus [1]. Other viruses such as the bovine leukemia virus [1,9], bovine viral diarrhea virus [1], and hog cholera virus can also be mechanically transmitted by tabanids; however, this is not the main route of infection for those pathogens [1]. There are reports of mechanical transmission of *Bacillus anthracis*, *Anaplasma marginale* (normally biologically transmitted by ticks), and *Francisella tularensis,* as well as some other bacterial pathogens. The protozoan pathogen *Besnoitia besnoiti* and many species in the *Trypanosoma* genus can also be mechanically transmitted by various species of tabanids [1].

Advances in sequencing technologies and bioinformatics have greatly expanded our knowledge of viral biodiversity. Thousands of new viruses have been discovered, mostly in arthropods [10,11,12]. Currently, viromes of well-established vector invertebrates, such as mosquitoes [13,14,15,16,17] and ixodid ticks [18,19,20] are being actively studied, while other blood-sucking invertebrates are receiving much less attention. To our knowledge, no specific work dedicated to description of tabanid viromes exists. However, the viromes of five unidentified specimens of tabanids (*Tabanidae* sp.) were uncovered during a large-scale insect virome study. As a result, five new viruses were discovered: the Wuhan horsefly virus, Jiujie fly virus, Wuhan horsefly virus 3, Hubei picorna-like virus 17, and Hubei toti-like virus 19 [11].

In this work, we explored the RNA viromes of several species in the *Hybomitra, Tabanus, Chrysops,* and *Haematopota* genera collected in Russia. Overall, we were able to identify and assemble fourteen full coding genomes of novel viruses, four partial coding genomes, as well as several fragmented viral sequences, which presumably belong to other twelve new viruses.

## 2. Materials and Methods

### 2.1. Collection and Pooling of Tabanids

Tabanids were collected manually in 2021 in the Primorye Territory and Ryazan Region, Russia. Tabanids were collected far from areas with large aggregations of livestock. Tabanid species were determined immediately after collection using taxonomy keys [21,22]. Species composition of high-throughput sequencing (HTS) pools, location collection, and date of material collection are presented in Table 1.

### 2.2. Sample Preparation and High-Throughput Sequencing

Tabanids were washed in 70% ethanol and then twice in distilled water prior homogenization. *Hybomitra* spp and *Tabanus* spp. were individually homogenized in 700 μL of saline solution and *Chrysops* spp. specimens were homogenized in 500 μL of saline solution (FSASI Chumakov FSC R&D IBP RAS, Moscow, Russia). Homogenization was carried out using Tissue Lyser II for 12 min at 25 s^−1^. Equal amounts of individual homogenates were pooled together on the basis of genera and collection site (Table 1).

RNA isolation, rRNA depletion, library preparation, and HTS were carried out as described previously [23]. All obtained raw reads were deposited in the sequence read archive (BioProject accession number PRJNA1026651).

### 2.3. High-Throughput Sequencing Assembly and Analysis

Raw high-throughput sequences were processed using Trimmomatic v0.39 [24], SPAdes v3.13.0 [25], and BLAST v2.9.0+ [26], as described previously [23].

In some cases, the obtained contigs themselves were additionally reassembled using SeqMan 7.0.0 (DNAstar Inc., Madison, WI, USA). After assembly, open reading frames were extracted from putative viral genome sequences and were tested using the blastp algorithm to detect virus related contigs. Some contigs with very high identities to known human pathogens (sequenced in the same run) were filtered out as possible contaminations.

We identified the closest relatives of each virus sequence using the online blastp algorithm. For each virus sequence with similar closest relative results, an estimation of evolutionary divergence was performed to assess whether they belong to the same virus species.

The abundance of viral reads in each pool was estimated using Bowtie 2 v.2.3.5.1 [27] software as described earlier [23]. Abundance of the largest contig containing tabanid 28S rRNA sequence was used as a positive control.

### 2.4. Phylogenetic Tree Construction

From the obtained contigs, we extracted either the polyprotein (if available) or RNA-dependent polymerase protein sequence. Homologs of the extracted sequence were extracted from the database performing online blastp searches with default parameters. The obtained sequences were filtered to remove sequences with low length, using custom Python script.

Subsequently, these sequences were aligned using MAFFT v7.310 [28] with E-INS-i algorithm and 1000 cycles of iterative refinement. Alignments were processed using the TrimAL v1.4. rev 15 program [29] in order to remove ambiguously aligned regions with automated region detection (“automated1” option). After that, sequences containing more than 10% of gaps or unknown amino acids were removed from alignments using custom Python script. Maximum-likelihood phylogenetic trees were constructed using the phyML 3.3.20200621 program [30] with 1000 bootstrap replications. Phylogenetic trees were annotated using custom Python script and visualized in FigTree v.1.4.4. The custom Python scripts used in this work are available at GitHub (https://github.com/justNo4b/slepni_scripts (accessed on 27 October 2023)).

### 2.5. Data Visualization

Phylogenetic tree visualization and image post-processing were performed as described previously [23]. The custom Python script for drawing genomes of the viruses is available at GitHub (https://github.com/justNo4b/GenomeDrawing (accessed on 27 October 2023)).

### 2.6. Virus Passages in Cell Lines

Three cell cultures were used in this work: a HAE/CTVM8 cell line [31], originating from *Hyalomma anatolicum* ticks; a C6/36 cell line, originated from *Aedes albopictus* mosquitoes; and a porcine embryo kidney (PEK) cell line. The PEK cell line was maintained at 37 °C in Medium 199 of Earle’s salts (FSASI Chumakov FSC R&D IBP RAS, Moscow, Russia), supplemented with 5% fetal bovine serum (Gibco, Paisley, UK). The C6/36 cell line was maintained at 32 °C in L-15 medium (FSASI Chumakov FSC R&D IBP RAS, Moscow, Russia), supplemented with 5% fetal bovine serum (Gibco, Paisley, UK). The HAE/CTVM8 cell line was maintained at 28 °C in L-15 medium, supplemented with 20% FBS, 10% Tryptose Phosphate Broth, 1% L-glutamine, and 2 µg/mL ciprofloxacin antibiotic.

Before cell infection, pooled tabanid homogenates were filtered via centrifugation for 15 min at 1500 rcf using Corning Costar Spin-X 0.45 µm centrifuge tube filters (Corning, NY, USA).

For the experiment on the C6/36 and PEK cell lines, cells were seeded in 96-well cell culture plates (SPL Life Sciences, Pocheon-si, Republic of Korea) and cultivated for one to two days. Then, cells were infected with either 30 μL of pools of tabanid homogenate, or 30 μL of the cultural fluid collected from the previous virus passage, before being incubated in the thermostat at 32 °C for the PEK cell line and at 28 °C for the C6/36 cell line for 6–7 days. Three passages were performed overall.

For the experiment on the HAE/CTVM8 cell line, cells were seeded in 96-well cell culture plates (SPL Life Sciences, Pocheon-si, Republic of Korea) and cultivated for seven days. Then, cells were infected with 30 μL of pools of tabanid homogenate and kept in the thermostat at 28 °C. Medium was changed at weekly intervals via removal and replacement of 150 μL.

An additional passaging experiment was performed on pool 7. For this experiment, C6/36 and PEK cells were seeded in flat-sided culture tubes (Nunc, Waltham, MA, USA) in 2.2 mL of growth medium and cultivated for two days. Then, cells were infected with either 200 μL of pools of tabanid homogenate or 200 μL of the cultural fluid collected from the previous virus passage, and incubated in the thermostat at 32 °C for PEK cell line and at 28 °C for C6/36 cell line for 6–7 days. Six passages were performed overall.

### 2.7. Virus Detection after Passages

In order to detect viruses during passages, oligonucleotide pairs were designed for each virus detected via high-throughput sequencing (Appendix A). For detection, total RNA was isolated from samples and reverse transcription was performed using random oligonucleotides, as described earlier [23]. Then, PCR was performed using cDNA, virus-specific oligonucleotides, and DreamTaq DNA polymerase (Thermo Fisher Scientific, Vilnius, Lithuania). The obtained PCR products were analyzed in agarose gel, with bands of target length being extracted from the gel. Then, they were purified, sequenced, and analyzed as described previously [23]. The sample was counted as positive for a virus if the results were confirmed via sequencing.

### 2.8. Virus Detection by qPCR

In order to estimate viral load during passages, TaqMan probes were designed for viruses that we were able to detect using PCR. Prior to the RNA isolation procedure, 1 μg of the PEK cells RNA and 2 × 10^4^ copies of poliovirus RNA were added to each sample as an internal control. Total RNA was than isolated from samples using TRI reagent LS (Sigma-Aldrich, St. Louis, MO, USA) in accordance with the manufacturer’s instructions. Reverse transcription was carried out with random oligonucleotides using MMLV Reverse Transcriptase (Evrogen JSC, Moscow, Russia) in accordance with the manufacturer’s instructions. Reverse transcription for the internal control was carried in a separate tube using PVR1 oligonucleotide (Appendix A).

qPCR was carried out using the R-412 qPCR reaction kit (Syntol, Moscow, Russia) in accordance with the manufacturer’s instructions. For each virus, a specific oligonucleotide pair and fluorescent probe were used (Appendix A). Samples were amplified in a C1000 Thermal Cycler (Bio-Rad, Hercules, CA, USA) using a CFX96 Real-Time System (Bio-Rad, Hercules, CA, USA) fluorescent detector. The obtained amplification data were analyzed using Bio-Rad CFX Manager v.3.1 (Bio-Rad, Hercules, CA, USA).

qPCR for the internal control sample was carried out using the same reaction kit and equipment. We employed poliovirus-specific oligonucleotides and a probe to do so (Appendix A).

## 3. Results

### 3.1. High-Throughput Sequencing

In this study, we processed ten pools of tabanids collected in 2021 from two distant regions of Russia: the Primorye Territory (Far East) and Ryazan Region (European part). Several species were studied from four genera: *Hybomitra*, *Chrysops*, *Tabanus*, and *Haematopota*.

We obtained 7–18 million reads per pool after filtration and managed to assemble 15 complete viral coding sequences (Figure 1, Table 2 and Appendix A). In four more cases, we were able to assemble partial coding sequences of the viruses (with gaps estimated to be less than 10% of the coding sequence). Additionally, we detected genome fragments that may indicate the presence of at least 12 more viruses in the studied samples (Figure 1, Table 2 and Appendix A).

All detected viruses, except for one, were significantly different from those already described in public databases and thus could be considered novel. All of them were close to various groups of the RNA viruses, including the Negevirus group, families *Narnaviridae*, *Totiviridae*, *Flaviviridae*, *Xinmoviridae*, *Permutotetraviridae*, *Dicistroviridae*, *Phasmaviridae*, *Solinviviridae*, *Rhabdoviridae*, *Iflaviridae*, *Noraviridae*, *Chuviridae*, and *Solemoviridae*.

The numbers of viruses in the samples varied greatly. No virus was detected in three pools (pools 4, 5, and 10), while ten were detected in pool 7 (*Chrysops relictus*) and six were detected in pools 6 and 8. The presence of viral reads was low overall, reaching a maximum at 2.68% in pool 7.

### 3.2. Negev-like Viruses

Negevirus is a genus proposed by Vasilakis and co-authors [32]. However, it is still officially unrecognized by the International Committee on Taxonomy of Viruses (ICTV). Negeviruses are characterized by single-stranded, positive-sense RNAs with poly(A) tails. The genomes of the viruses range in size from 9 to 10 kb and encode three overlapped open reading frames (ORFs). All negeviruses were isolated from mosquitoes and phlebotomine sand flies [32]. Recently, several similar negev-like viruses were discovered in various insects during virome studies. Many of these newly discovered viruses have a longer genome size and up to five non-overlapping ORFs [33,34].

Here, we report the discovery of three negev-like contigs in our study. These were preset in pool 1 (*Hybomitra brevis*), pool 6 (*Haematopota pluvialis*), and pool 7 (*C. relictus*). These contigs were 11.6–11.9 kb in length and contained five ORFs with an overall layout similar to that of negev-like viruses (Figure 2B). According to BLAST assessments, all the contigs had quite low similarity (39.2–44.3%) to their closest relatives. Comparison of contigs with each other using a blast program showed that they are only distantly related with each other (68.9–71.2% identity with 51–68% of query cover), suggesting that each contig represents a separate novel negev-like virus (Appendix A). The viruses were named Xanka Hybomitra negev-like virus (XHNV), Melisia Haematopota negev-like virus (MelHNV), and Medvezhye Chrysops negev-like virus (MedCNV). All of these viruses had a relatively low abundance in the pools, accounting for 0.01%, 0.23%, and 1.08% of the total reads, respectively.

Phylogenetic analysis showed that all these viruses formed a single well-supported group (Figure 2A), with a sister relationship to a clade formed by viruses of fruit flies from the genera *Zeugodacus*, *Ceratitis*, and *Bactocera* [34]. It should be noted that, within the tabanid clade, MedCNV and MelHNV, both of which were found in tabanids in the Ryazan region, form a well-supported group. However, tabanid phylogenetic trees placed the genera *Haematopota* and *Hybomitra* closer to each other than to the genus *Chrysops* [3]. Such a situation hints towards a geographically driven evolution of tabanid negev-like viruses.

### 3.3. Flavi-like Virus

Classical *Flaviviridae* members are small, enveloped viruses with positive-sense RNA genomes. They are generally 9–13 kb in length. All members lack poly-A tails, and only members of the genus *Orthoflavivirus* contain a cap structure. Others instead possess an internal ribosomal entry site. All members encode a single ORF that is processed by viral and cellular proteases into several structural and non-structural proteins. Non-structural proteins contain regions encoding a serine protease, RNA helicase, and RNA-dependent RNA polymerase, and the order of these domains is conservative within the family [35].

Recently, several groups of viruses with homology to the *Flaviviridae* polymerases were discovered, and some of them had segmented genomes. One of these groups contained viruses with huge monopartite RNA genomes, up to 30 kb in length [10,11]. Those viruses were found in insects, ticks, and even plants [10,11,36,37].

In this work, we discovered a 20.9 kb contig with homology to the *Flaviviridae* polymerase in pool 6 (*Ha. pluvialis*). The contig contained a single ORF 6715 aa in length, flanked by untranslated regions on the 5′ and 3′ ends (Figure 3B). According to BLAST analysis, the contig had a 41.7% identity with 15% cover to the Orthopteran flavi-related virus. Thus, it represents a novel flavi-like virus, and it was named Medvezhye Haematopota flavi-like virus (MHFV).

MHFV had 0.29% abundance in the pool. Phylogenetic analysis showed (Figure 3A) that MHFV groups together with the Xingshan cricket virus [37] (with <70% bootstrap support). Other close relatives (with <70% bootstrap support) include viruses of the *Culex* mosquitoes (Placeda virus [38], Culex tritaeniorhynchus flavi-like virus [33]) and *Musca domestica* (Shayang fly virus 4 [37]).

### 3.4. Xinmo-like Virus

The family known as *Xinmoviridae* contains single negative-strand RNA viruses 9–14 kb in length, encoding three to six proteins. Viruses within this family have mostly been discovered using HTS in various species of insects, including mosquitoes, parasitoid wasps, flies, dragonflies, and others. The taxonomy of *Xinmoviridae* has recently been revised, with several new genera being created [39].

Here, we found several contigs with homology to the *Xinmoviridae* proteins (Figure 4B). The first one was found in pool 6 (*Ha. pluvialis*) and was 11.5 kb in length. It encoded four ORFs with a typical *Xinmoviridae* layout. According to BLAST analysis, the closest relative was Hubei diptera virus 11 (*Alasvirus muscae*) with 35.9% aa identity in the polymerase (98% query cover). Such a low identity across the polymerase shows that this contig represents the genome of a novel virus, which was named Medvezhye Haematopota xinmo-like virus (MHXV).

In mixed pool 8 (*C. pictus/C. caecutiens*), we found two contigs encoding *Xinmoviridae*-related proteins. One of them encoded four proteins, including partial polymerase, and the second one encoded approximately 70–80% of the polymerase. After performing a protein alignment of the partial fragments of this contig on MHXV polymerase, we concluded that they are likely to belong to the single virus named Medvezhye Chrysops xinmo-like virus (MCXV), with a gap of about 500 nt in the polymerase region. This virus polymerase had a 60.1% amino acid identity with the MHXV polymerase (Appendix A) and a 39.2% identity to the closest relative found in Genbank (98% query cover).

Additionally, in pool 6 (*Ha. pluvialis*), there was a small 580 nt contig that encoded a part of a *Xinmoviridae*-like polymerase; however, it was distant to both MHXV and MCXV (Appendix A). The working name of this genome fragment is given in Appendix A.

The abundance of MHXV in the pool was 0.18%, while that of MCXV was 0.01%. Phylogenetic analysis showed that MHXV and MCXV form a monophyletic group in the *Xinmoviridae* family polymerase tree (Figure 4A), with the closest relatives being viruses of diptera (Hubei diptera virus 11 [11], Shuangao fly virus 2 [12], and Gudgenby Calliphora mononega-like virus [40]) and wasps (Hymenopteran anphe-related viruses OKIAV72 and OKIAV71 [41]).

During revisions of *Xinmoviridae* taxonomy, criteria were introduced for new genera and species. According to the accepted proposal, a new virus species must have a near-complete coding genome and an RdRp amino acid identity of 66% or lower, while a novel *Xinmoviridae* genus should have an RdRp amino acid identity lower than 60% [39].

MCXV does not qualify as a novel virus species due to the large gap in the polymerase sequence. MHXV has a 35.9% identity in the polymerase to the closest relative and has a complete coding genome determined. Thus, MHXV may qualify as a novel *Xinmoviridae* genus according to those guidelines.

### 3.5. Toti-like Viruses

Classical members of *Totiviridae* contain single-segment double-stranded RNA genomes of 4.5–7 kbp in length, with two often overlapping ORFs. The first ORF encodes a major capsid protein (CP), and the second one encodes RdRp. Classical totiviruses infect eukaryotic microorganisms, such as *Leishmania* spp. and *Trichomonas* spp., or fungi [42]. However, the results of recent metagenomic studies show a large diversity of toti-like viruses in insects [11].

In the current study, we were able to find several contigs containing *Totiviridae*—like ORFs (Figure 5B). In pool 2 (*Hybomitra nigricornis*), we found a single 5621 nt contig with two distinct ORFs that had homology to the *Totiviridae* CP and RdRp. According to RdRp BLAST, it had a 41.1% identity to the closest relative (Bactrocera zonata toti-like virus). Such a low identity across the polymerase shows that this contig represents a genome of the novel virus, and it was named Volxa Hybomitra toti-like virus (VHTV). VHTV had 0.01% abundance in the pool.

In pool 8 (*C. pictus/C. caecutiens*), we were able to detect a 7348 nt contig containing two ORFs with homology to the *Totiviridae* CP and RdRp. According to RdRp BLAST, there was a 60% similarity to the closest relative (Hubei toti-like virus 19); thus, we considered it a novel virus and named it Medvezhye Chrysops toti-like virus (MCTV). The abundance of MCTV in the pool was 0.03%.

Additionally, in pool 9 (T. *autumnalis/T. bromius*), we detected four contigs with homology to the *Totiviridae* proteins; however, we were not able to assemble a complete genome from them. The contigs had a 67–79% identity to the CP and RdRp of Hubei toti-like virus 19 and 54.2–67.9% identity to MCTV. Thus, we can speculate that all of them belong to the genome of a single virus, which is more closely related to Hubei toti-like virus 19 than to MCTV.

Phylogenetically, MCTV and VHTV belong to two unrelated groups of toti-like viruses (Figure 5A). VHTV seems to be close (although with <70% bootstrap support) to the viruses of *Bactrocera* fruit flies (Bactrocera dorsalis toti-like virus 1 [43] and Bactrocera zonata toti-like virus Bz-V4 [34]). MCTV formed a monophyletic group with Hubei toti-like virus 19 [11] isolated from unspecified species of tabanids. Other relatives include viruses found in wasps, ants, and soldier flies, while viruses of fruit flies form a separate monophyletic group.

### 3.6. Narna-like Viruses

Classical members of the *Narnaviridae* family are capsidless viruses that possess a positive-strand RNA genome, 2.3–2.9 kb in length. The genome encodes a single viral protein, RdRp. Classical *Narnaviridae* members infect fungi. Recently, a vast number of narna-like viruses were found in insects [44]. While some of these follow the classical *Narnaviridae* genome plan, there are data indicating that at least some of them encode a second functional ORF on the minus strand of the genome [45,46].

Here, we detected three narna-like contigs (Figure 6B,C). The first one was detected in pool 2 (*H. nigricornis*) and was 2208 nt in length. It contained a single ORF with homology to the narna-like RdRp and, according to BLAST, had 52.5% identity to the closest relative. Thus, we considered this contig a genome of a novel narna-like virus and named it Kamenushka Hybomitra narna-like virus (KHNV). KHNV had an extremely low presence in the pool (less than 0.01%).

In pool 7 (*C. relictus*), we found two narna-like contigs. They were similar in length (2423 and 2342 nt) and had two ORFs, with one of them encoding RdRp. According to RdRp BLAST, both had around 51% identity to the RdRp of Sanya cydistomyia duplonotatay narnavirus 1. When compared with each other, the polymerase-encoding ORFs of the two detected contigs had 87.2% identity. Thus, we considered those two contigs two separate novel narna-like viruses and named them Medvezhye Chrysops narna-like virus 1 (MCNV1) and Medvezhye Chrysops narna-like virus 2 (MCNV2). The abundance of MCNV1 and MCNV2 in the pool was 0.61% and 0.23%, respectively.

Phylogenetically, KHNV formed a monophyletic group with an RdRp found in the metagenome of *Parus caudatus* (insectivorous bird) with no other close relatives (Figure 6A). MCNV1 and MCNV2 formed a monophyletic group with Sanya cydistomyia duplonotatay narnavirus 1 (found in the *Sanya cydistomyia* tabanid) and Hubei narna-like virus 20 (found in unspecified Diptera) [11].

### 3.7. Solemo-like Viruses

The members of the *Solemoviridae* family are non-enveloped plant viruses with a ~4–6 kb positive-sense RNA genome. They use mechanisms such as leaky scanning, subgenomic RNA production, and ribosomal frameshifting to express viral proteins [47]. Recently, many new viruses with RdRps related to *Solemoviridae* were discovered. They are mostly found in insects and can drastically differ in their overall genome structure. Some of them have different ORF counts and/or have a genome divided into two segments [11].

In this work, we discovered a number of *Solemoviridae*-related contigs (Figure 7B,C). In pool 2 (*H. nigricornis*), we found four contigs. We identified one of them as a full second segment with 52.7% identity to the closest GenBank relative. We were able to assemble three remaining fragments into a partial sequence of the first segment (with only 5 nt remaining unknown), using its closest GenBank relative (Ulaatai Melophagus solemo-like virus [23]) as a reference. Overall, the first segment had a 63.6% identity in the RdRp. Therefore, we considered those two contigs to constitute a genome of a single solemo-like virus and named it Komarovka Hybomitra solemo-like virus (KHSV). Overall, KHSV had less than 0.01% abundance in the studied pool.

In pool 8 (*C. pictus/C. caecutiens*), we identified two solemo-related contigs. One of them contained a full sequence with an ORF layout typical for a segmented solemo-like virus and had a 46.6% identity to the closest GenBank relative in the RdRp. The second contig, according to BLAST, had a 42.2% identity to the closest relative and contained a partial sequence of the second segment, with approximately 10–15 aa missing on the 5′ side of the VP3 ORF. Therefore, we considered these two contigs as the genome of a single solemo-like virus and named it Istie Chrysops solemo-like virus (ICSV). ICSV abundance in the studied pool was 0.01%.

In pool 7 (*C. relictus*), we found several contigs with homology to solemo-like proteins. Three contigs had a homology to the RdRp of the different viruses (39.5–55.7% identity), and all of them had a typical segmented solemo-like ORF layout. Other three contigs had an ORF layout typical for the second segment of the solemo-like viruses and homology CP (35.4–52.6% identity). Therefore, we considered these six contigs to be genomes of three novel solemo-like viruses and named them Medvezhye Chrysops solemo-like virus (MCSV), Melisia Chrysops solemo-like virus (MelCSV), and Polka Chrysops solemo-like virus (PCSV). MelCSV had the highest abundance (0.28%) in the studied pool, while the abundance of MCSV and PCSV was lower, standing at 0.13% and 0.05%, respectively. The first and second segments were grouped together only on the basis of their homology to similar viruses found in the GenBank.

Phylogenetic analysis based on the polymerase sequence showed that the identified viruses are divided into three distinct groups (Figure 7A). MelCSV and KHSV formed a monophyletic group related to the viruses of odonata (Hubei diptera virus 14 [11]), birds (*Riboviria* sp. viruses), and *Melophagus ovinus* (Ulaatai Melophagus solemo-like virus [23]). PCSV and ICSV formed another monophyletic group related (although with <70% bootstrap support) to other Diptera viruses, including viruses of *Drosophila* (Teise virus, Motts Mill virus [48,49]), *Musca vetustissima* (Jeffords solemo-like virus [40]), and *M. ovinus* (Bayan-Khairhan-Ula Melophagus solemo-like virus [23]). MCSV formed a separate branch, with low bootstrap support to any proposed groupings.

### 3.8. Permutotetra-like Virus

The *Permutotetraviridae* family contains the single genus *Alphapermutotetravirus*, with two member species that infect Lepidopteran insects. They are characterized by a monopartite single-stranded (+) RNA genome, containing two overlapping ORFs. The first ORF encodes a unique internally permuted polymerase, with a C–A–B arrangement of the canonical motifs found in the palm subdomain of all polymerases. The second ORF encodes a capsid protein and is expressed from a subgenomic RNA [50]. Recent metagenomic advancements in virology resulted in the discovery of many new permuto-like viruses [11].

In this work, we discovered a 4.6 kb contig with homology to the permuto-like polymerase in pool 6 (*Ha. pluvialis*). Further analysis showed that it had two ORFs in the typical permuto-like order: the first one encoded a polyprotein with the RdRp domain, and the second one encoded CP (Figure 8B). According to the BLAST analysis of the polyprotein ORF, there was only a 35.8% identity to the closest relative with 61% query cover. Such a low identity across the polyprotein shows that this contig represents the genome of a novel virus, which was named Medvezhye pound Haematopota permuto-like virus (MHPV). MHPV had 0.03% abundance in the pool. Phylogenetically, MHPV is a sister group to various permuto-like viruses of insects (Figure 8A).

### 3.9. Big Sioux River Virus

Big Sioux River virus (BSRV) is a dicistro-like virus. It has a positive-strand RNA genome of 10 kb with two ORFs, typical for *Dicistroviridae* members. The first ORF encodes nonstructural proteins, including RdRp, and the second one encodes capsid proteins. BSRV was first isolated from honeybees (*Apis mellifera*) [51], and later detected in soybean aphids (*Aphis glycines*) and *Culex tritaeniorhynchus* mosquitoes in China [16] and *Aphis fabae* in Kenya [52].

We detected a 10,271 nt long contig in pool 7 (*C. relictus*). It had two ORFs, typical for *Dicistroviridae* (Figure 9B), as well as very high homology to BSRV (99% identity with 99% query cover) in the RdRp-encoding ORF. Thus, we considered this contig the genome of a novel BSRV strain and called it Medvezhye strain. Phylogenetically, our strain formed a clear monophyletic group with all other BSRV strains, except QGX47955, which clustered together with several isolates of the Aphis gossypii virus (Figure 9A).

### 3.10. Ifla-like Viruses

Classical members of the *Iflaviridae* family are non-enveloped, single-stranded, non-segmented, and positive-sense RNA viruses. Their genomes are 9–11 kb in length and encode a single ORF. This ORF encodes a polyprotein that is processed into several virus proteins, including RdRp. All members of the *Iflaviridae* family have been isolated from arthropods [53]. Recently, the diversity of iflaviruses has expanded significantly due to the study of insect viromes using HTS [11].

In the current work, we managed to find several ifla-related contigs in the studied material. The first one, found in pool 8 (*C. pictus*/*C. caecutiens*), was 8719 nt long and had a single ORF (Figure 10B). According to BLAST analysis of this ORF, it had 34.5% identity to the closest relative (Exitianus exitiosus virus 2); however, the contig was truncated compared to a full ifla-like genome, lacking 5′UTR and a small 5′-terminal part of the polyprotein encoding sequence. Thus, we considered this contig to be the partial genome of a novel ifla-like virus and named it Medvezhye Chrysops ifla-like virus (MCIV). MCIV had a 0.02% presence in the pool.

In pool 2 (*H. nigricornis*), we found seven contigs related to various ifla-like viruses; however, we were not able to assemble them in a single genome. The contigs varied in length (408–1234 nt) and had 31.8–63% identity to the closest relative according to online BLAST. At the same time, those contigs showed higher identity (Appendix A) to the MCIV polyprotein (35–72%). Additionally, we found a single 381 nt contig in pool 3 (*Hybomitra stigmoptera*), with homology to ifla-like protein. Interestingly, the ORF this contig encoded was closer to the MCIV polyprotein (79.9% identity) than to the NODE_21_length_616 contig from the *H. nigricornis* pool (52% identity) and to the closest GenBank entry (39.6%). Thus, we assume that at least one ifla-like virus might exist in the discussed *Hybomitra* pools. The working names of the genome fragments are given in Appendix A.

Phylogenetically, MCIV forms a monophyletic group (Figure 10A) with ifla-like viruses of various insects (Shuangao insect virus 12 [11]), including leafhoppers (Exitianus exitiosus virus 2 [54]), mantis fly (Sanya iflavirus 8), alfalfa weevil beetle (Hypera postica associated iflavirus 2 [55]), and the common wasp (Leuven wasp-associated virus 5 [56]). Although we were not able to construct a reliable phylogenetic tree using genome fragments found in *H. nigricornis* and *H. stigmoptera*, we can speculate that they are likely to group in a monophyletic group with MCIV due to its higher polyprotein identity compared to any GenBank entry.

### 3.11. Virus-like Fragments

In addition to the abovementioned toti-like, ifla-like, and xinmo-like genome fragments, we also managed to detect contigs related to genome fragments of the following virus groups: *Orthophasmavirus* (*Bunyavirales*, *Phasmaviridae*), Nora virus (*Picornavirales*, *Noraviridae*), *Solinviviridae* (*Picornavirales*), *Chuviridae* (*Jingchuvirales*), and *Rhabdoviridae* (*Mononegavirales*). The working names of the genome fragments are given in Appendix A.

Solinviviridae-like contigs were detected in both pool 7 (*C. relictus*) and pool 8 (*C. pictus/C. caecutiens*). In both cases, amino acid sequences of the fragments showed similarity (38–82% identity for different fragments) to the polyprotein of Hangzhou solinvi-like virus 2 (Appendix A), which was found in the *Orthetrum testaceum* dragonfly metagenome. However, contigs from both pools had very high levels of similarity (96.9–99.5% identity), even on the nucleotide level, and the abundance of reads in pool 8 was very low (Appendix A). We believe that the possibility of read contamination during the sequencing run in the case of pool 8 is likely, and do not consider reads in pool 8 as a detected virus.

There was a single 616 nt Nora-virus-like contig found in pool 6 (*Ha. pluvialis*). Its closest relative (42.9% aa identity) was Caledonia beadlet anemone nora virus-like virus 1, found in *Actinia equine*, a common sea anemone.

Orthophasma-like contigs were found in three pools: pool 1 (*H. brevis*), pool 8 (*C. pictus/C. caecutiens*), and pool 9 (*T. autumnalis/T. bromius*). In the *C. pictus/C. caecutiens* pool, we were able to assemble sequences that represent a full coding sequence of the glycoprotein, and about 80% of the RdRp and nucleocapsid protein. In the other two pools, contigs were significantly smaller. Overall, contigs encoded ORFs related to all three *Orthophasmavirus* segments, with amino acid identity varying from 31.5 to 69.3% and the closest relative in most cases being Tibet bird virus 1, detected in the bird feces metagenome. Detected contigs showed 42–77.3% identity in a pool-to-pool comparison, implying that contigs from different pools belong to different viruses (Appendix A).

In pool 7 (*C. relictus*), three contigs encoding rhabdo-like RdRp were detected. Further investigation showed that one of them was related to Wuhan fly virus 3 and Shayang fly virus 3 (*Rhabdoviridae*, *Alphadrosrhavirus*), while the other two contigs were related to Hubei lepidoptera virus 2 (*Rhabdoviridae*, *Alphapaprhavirus*). This shows that there may be at least two different rhabdo-like viruses in the *C. relictus* pool.

In pool 9 (*T. autumnalis*/*T. bromius*), we detected five contigs with homology to *Chuviridae* RdRp and glycoprotein. Most of the fragments were related to megalopteran chu-related virus 119, with 32.4–52.8% identity.

### 3.12. Virus Isolation

In addition to high-throughput sequencing, we performed research on the ability of the discovered viruses to reproduce in three cell cultures: C6/36 originating from *Aedes albopictus*; HAE/CTVM8 originating from *Hyalomma anatolicum* ticks; and a pig embryo kidney (PEK) cell line. Cell lines were infected with pools of the tabanid homogenate, as described in Table 1. For the C6/36 and PEK cell lines, three blind passages were performed. In the case of the HAE/CTVM8 cell line, three weeks of persistent infection with weekly changes of the culture medium were performed. After this, we tested the collected supernate using virus-specific oligonucleotides. πusing Sanger sequencing of the obtained PCR product. Thirteen viruses were detected in the HAE/CTVM8 cell culture after three weeks of persistence, eleven viruses were detected in the C6/36 cell culture, and nine viruses were detected in the PEK cell culture. Overall, seventeen viruses were detected using PCR (Table 3).

Since the number of viruses detected was particularly big, we decided to better estimate their replication dynamic by quantifying the number of viruses on each passaging step using qPCR with TaqMan probes. qPCR analysis was performed for all PCR-positive viruses, except MCIV, MedCNV, and MCSV. The results are presented in Appendix A.

No cases where the viral load consistently increased were observed. Overall, in the majority of cases, the viral load decreased with each passage (PEK and C6/36 cell lines) or week of chronic infection (HAE/CTVM8 cell line). In some cases, we observed the virus load increase between the first and second weeks of chronic infection in the HAE/CTVM8 cell line (Barsukovka Hybomitra ifla-like virus, MCNV2, PCSV, MelCSV, BSRV, and VHTV). We also observed the virus load increase between the first and second passage in the C6/36 cell line in the case of VHTV. Additionally, the virus load increased between the second and third weeks of chronic infection for Medvezhye Tabanus toti-like virus in HAE/CTVM8 and between the second and third passage in PEK cell line. The same could be said for ICSV in the HAE/CTVM8 cell line.

The third passage in the PEK cell line was additionally tested for the presence of viruses using HTS. Overall, genome fragments of eight viruses were detected (Table 4). Five of them were also detected via PCR (Table 3), while three of them were detected only via HTS. It should be noticed, however, that only a small number of reads were detected and we were not able to assemble a full coding genome for any detected virus after three passages in the PEK cell line.

Additionally, we performed six passages of the pool 7 (*C. relictus*) in the PEK and C6/36 cell lines. However, no viruses were detected using virus-specific PCR oligonucleotides in either culture.

## 4. Discussion

Viromes of some blood-sucking ectoparasites, such as mosquitoes and ticks, have been relatively well studied [13,14,15,16,18,19,20] due to their importance as vectors for various human and animal pathogens. Viromes of tabanids, the largest group of hematophagous insects [2,3], remain relatively unstudied. Here, we present data on the viromes of several species in the *Hybomitra*, *Tabanus*, *Chrysops*, and *Haematopota* genera collected in Russia. Previously, the viromes of five unidentified specimens of tabanids were studied using HTS, and five novel viruses were discovered [11].

Here, we explored the RNA viromes of several species of tabanids collected in different parts of Russia, namely in the Primorye Territory and Ryazan Region. In our study, different pools contained from 0 to 10 viruses, with about three viruses per pool on average. Virus diversity was higher in the samples collected in the Ryazan Region (Figure 1). Currently, with different species of tabanids analyzed and collected in places that are far from each other and at different times, it is hard to determine whether this difference is a result of territory, tabanid species, or some other factors.

ICTV species determination guidelines differ for different genera [39,57], and in many cases, they are not actively used for novel viruses discovered via HTS [35,47,53]. At the same time, ICTV requires uncultivated virus genome sequences to have at least a complete coding sequence in order to be accepted for taxonomic classification [58]. Here, for simplicity, we used one universal criterion for the determination of novel viruses, namely, that they should have less than 90% similar amino acid identity in the polymerase-encoding ORF to their closest relative. We also proposed working names for all detected viruses. Obviously, the final decision on their nomenclature and classification can only be made by the ICTV.

Overall, we were able to identify 30 novel viruses. Fourteen of them had complete coding sequences assembled, and thus can be accepted for further taxonomic classification [58]. Overall, the identified viruses included positive-sense and negative-sense RNA viruses from 12 distinct groups. Only one virus among these had previously been described, while the others differed significantly from the viruses found in the GenBank database. Moreover, MHXV even qualified for genus demarcation criteria in the *Xinmoviridae* family according to the ICTV guidelines [39]. Thus, our work contributes to the description of virus biodiversity.

In most cases, newly discovered viruses clustered together with each other, other viruses of tabanids, or members of Diptera. In the case of toti-like, solemo-like, and narna-like viruses, several clusters of tabanid viruses were observed across the phylogenetic trees. Previously, a similar situation had been observed for solemo-like viruses of *M. ovinus* [23]. Interestingly, in several cases (xinmo-like viruses, ifla-like viruses, and one group of toti-like viruses), viruses found in wasps were closely related to tabanid viruses, with mosquito viruses, for example, being more distant. While this situation may be the consequence of our lack of knowledge regarding insect viruses, it may also be a sign of virus interspecies transmission due to ecological interactions. For example, sand wasps can prey on various species of tabanids [5,59]. Overall, additional information about insect viromes will improve our understanding of virus speciation.

There are several viruses that are known to be mechanically transmitted by tabanids [1,9]. Here, we did not find any of those viruses, or any viruses that may be considered their close relatives. This was an expected result, since we collected tabanids far from areas with large aggregations of livestock.

BSRV was the only known virus that was found in the pools. This was the first recorded detection of BSRV in tabanids. It had previously been detected in honeybees (*A. mellifera*) [51], and later in the soybean aphid (*A. glycines*) and *C. tritaeniorhynchus* mosquitoes in China [16], as well as in *Ap. fabae* in Kenya [52]. It should be noted, however, that honeybee BSRV had only 87% similarity with both strains isolated from *C. tritaeniorhynchus*, as well as with a strain isolated from *Ap. fabae* [52].

Here, we attempted to isolate novel viruses using three different cell cultures. Seventeen viruses were detected on the third passage; however, the virus load in qPCR decreased in the majority of cases. No cases were observed where the viral load consistently increased. These data indicate that, in the majority of cases, even if viruses are able to replicate in cell cultures, their reproduction level is very low and the viruses are slowly eliminated from cell cultures.

In several cases, the viral load increased between the first and second or the second and third passage. Those cases may be a sign of ongoing adaptation of the viruses to the cell cultures, and further work is needed in order to determine if they are able to reproduce stably or will be ultimately eliminated from the cell cultures. It should be noted that, in the majority of cases, an increase in viral load was observed during persistence in the HAE/CTVM8 cell line. The question of whether this is a result of the properties of this specific cell culture, or a result of a virus’s persistence cultivation scheme (passages were used with the C6/36 and PEK cell lines), should be the subject of future research.

## 5. Conclusions

We explored the RNA viromes of several species of tabanids collected in Russia. Full coding sequences of fourteen novel viruses were assembled. In four more cases, we were able to assemble partial coding sequences of the viruses and detected genome fragments that may indicate the presence of at least 12 more viruses in the studied samples. Thus, our work contributes to the description of virus biodiversity.

All detected viruses were studied for their ability to replicate in the C6/36, HAE/CTVM8, and PEK cell lines. While seventeen viruses were detected using PCR on the third passage (for PEK and C6/36 cell lines) or in the third week of chronic infection (HAE/CTVM8), the viral load steadily decreased in the majority of cases. No cases were observed where the viral load consistently increased. It seems that three passages are insufficient to conclude the isolation of viruses.

## Figures and Tables

**Figure 1 viruses-15-02368-f001:**
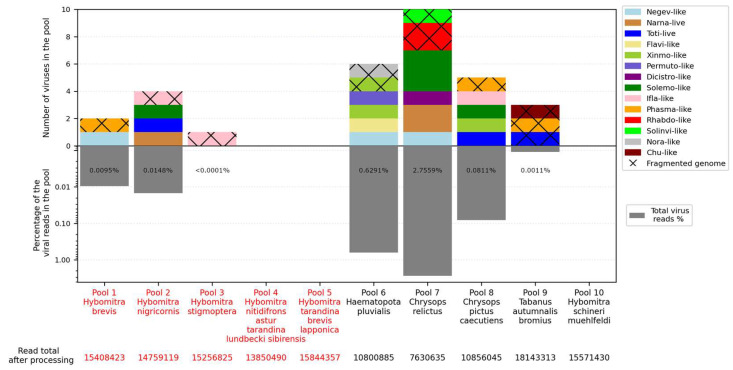
Abundance of viruses (**top**) and virus-containing reads (**bottom**) in each studied pool. Distinct virus groups are marked by color. Sections with only individual fragments of virus genome obtained are marked by crosses. Pools of tabanids from Primorye Territory and Ryazan Region are indicated in red and black, respectively, in the *X*-axis caption.

**Figure 2 viruses-15-02368-f002:**
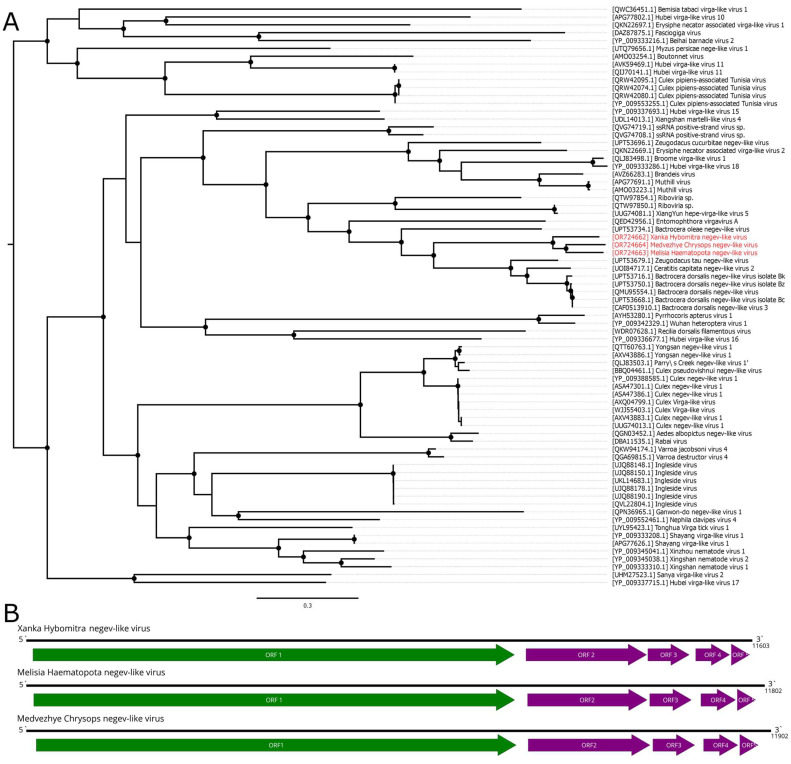
Genomic structure and phylogenetic relationships of the negev-like viruses described in this study. (**A**) Midpoint-rooted phylogenetic tree constructed using amino acid sequences of the ORF1 polyprotein (1000 bootstrap replicates; nodes with >70% bootstrap support are marked). Scale bar represents the number of amino acid substitutions per site. Discovered viruses are marked in red. (**B**) Scheme of the negev-like viruses’ genomes. RdRp-encoding ORF is marked in green. All other ORFs are marked in purple.

**Figure 3 viruses-15-02368-f003:**
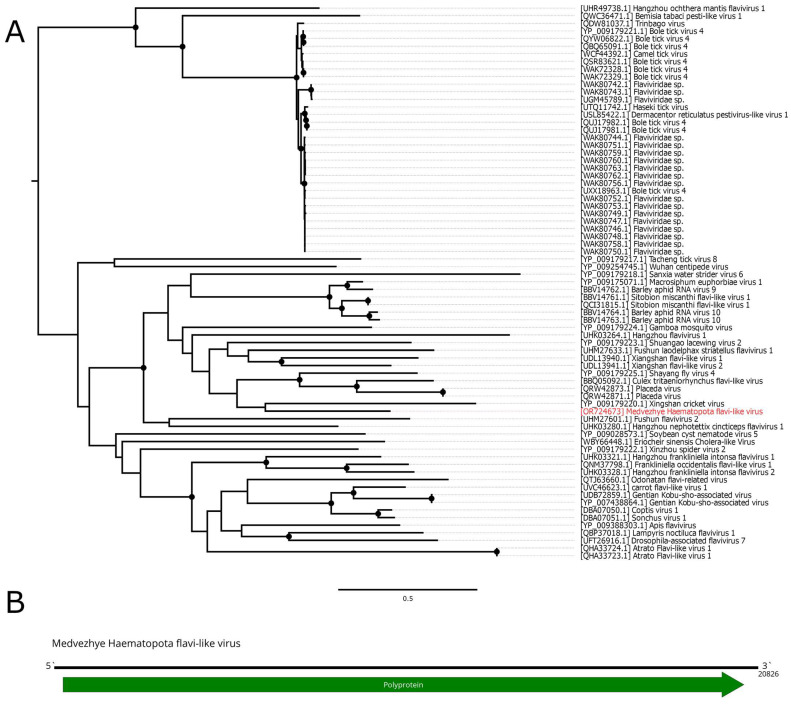
Genomic structure and phylogenetic relationships of Medvezhye Haematopota flavi-like virus. (**A**) Midpoint-rooted phylogenetic tree constructed using amino acid sequences of the polyprotein (1000 bootstrap replicates; nodes with >70% bootstrap support are marked). Scale bar represents the number of amino acid substitutions per site. Discovered viruses are marked in red. (**B**) Scheme of the Medvezhye Haematopota flavi-like virus genome. RdRp-encoding ORF is marked in green.

**Figure 4 viruses-15-02368-f004:**
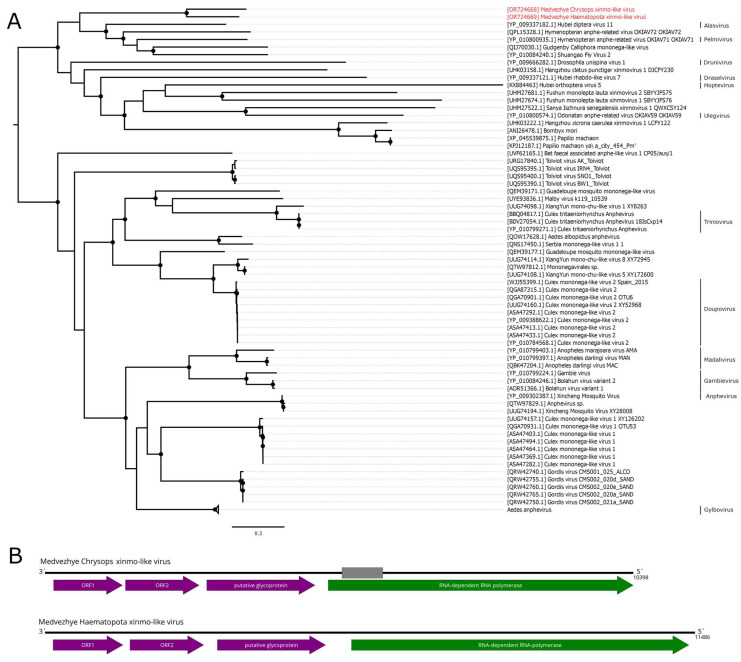
Genomic structure and phylogenetic relationships of xinmo-like viruses. (**A**) Midpoint-rooted phylogenetic tree of the *Xinmoviridae* family and related viruses constructed using amino acid sequences of the RdRp (1000 bootstrap replicates; nodes with >70% bootstrap support are marked). Scale bar represents the number of amino acid substitutions per site. Discovered viruses are marked in red. (**B**) Scheme of the xinmo-like viruses’ genomes. Grey bar on the genome represents a gap in the sequence. RdRp-encoding ORF is marked in green. All other ORFs are marked in purple.

**Figure 5 viruses-15-02368-f005:**
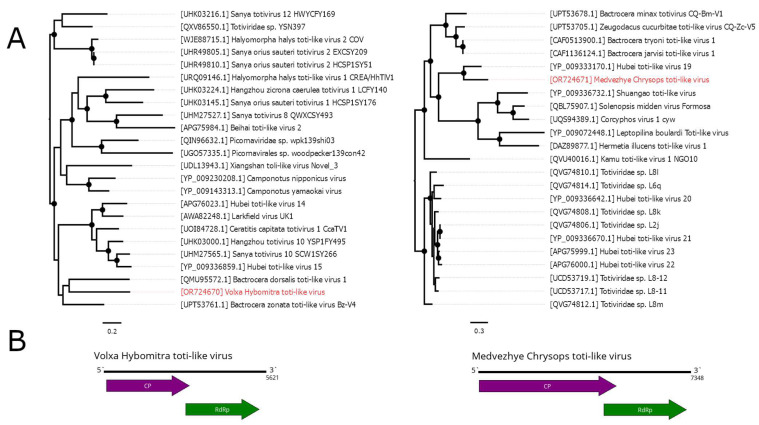
Genomic structure and phylogenetic relationships of the toti-like viruses. (**A**) Midpoint-rooted phylogenetic tree constructed using amino acid sequences of RdRp (1000 bootstrap replicates; nodes with >70% bootstrap support are marked). Scale bar represents the number of amino acid substitutions per site. Discovered viruses are marked in red. (**B**) Scheme of Volxa Hybomitra toti-like virus and Medvezhye Chrysops toti-like virus genomes. RdRp-encoding ORF is marked in green. All other ORFs are marked in purple.

**Figure 6 viruses-15-02368-f006:**
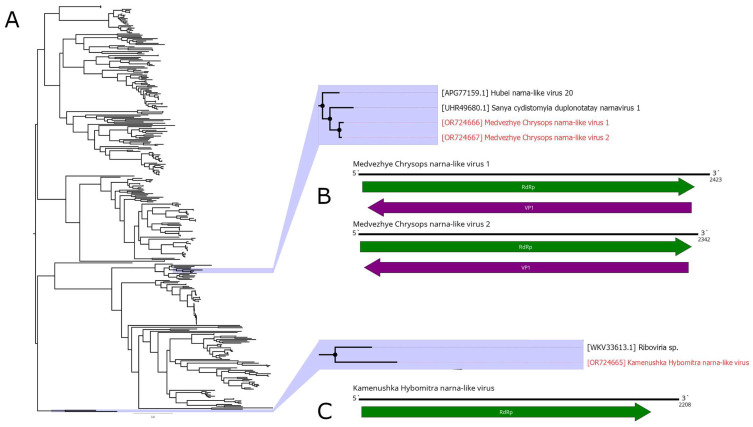
Genomic structure and phylogenetic relationships of narna-like viruses. (**A**) Midpoint-rooted phylogenetic tree constructed using amino acid sequences of the RdRp (1000 bootstrap replicates; nodes with >70% bootstrap support are marked). Scale bar represents the number of amino acid substitutions per site. Discovered viruses are marked in red. (**B**,**C**) Scheme of the narna-like viruses’ genomes. RdRp-encoding ORF is marked in green. All other ORFs are marked in purple.

**Figure 7 viruses-15-02368-f007:**
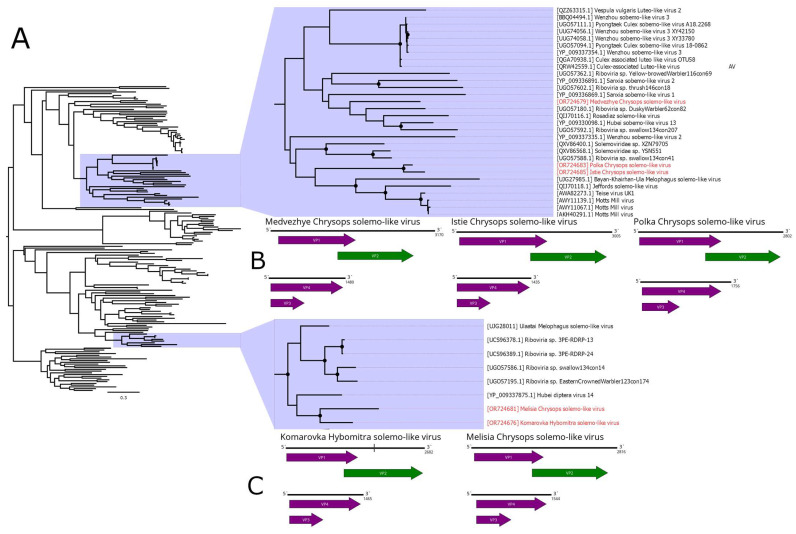
Genomic structure and phylogenetic relationships of solemo-like viruses. (**A**) Midpoint-rooted phylogenetic tree constructed using amino acid sequences of the VP2 protein (1000 bootstrap replicates; nodes with >70% bootstrap support are marked). Scale bar represents the number of amino acid substitutions per site. Discovered viruses are marked in red. (**B**,**C**) Scheme of solemo-like virus genomes. RdRp-encoding ORF is marked in green. The gap in Komarovka Hybomitra solemo-like virus is marked on its genome. All other ORFs are marked in purple.

**Figure 8 viruses-15-02368-f008:**
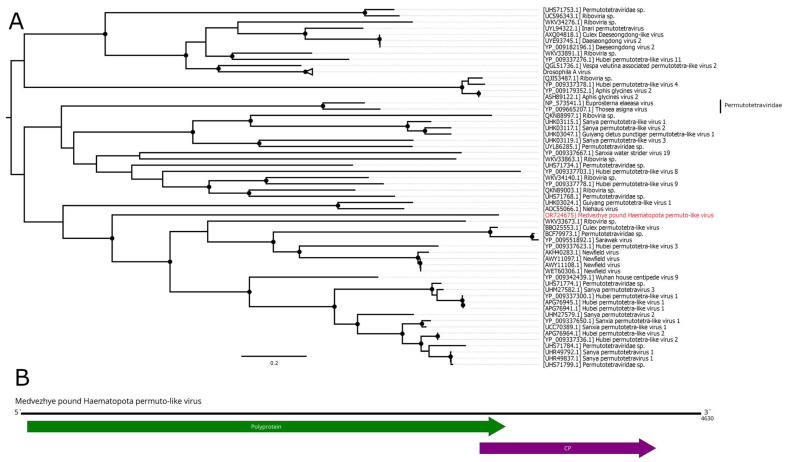
Genomic structure and phylogenetic relationships of Medvezhye pound Haematopota permuto-like virus. (**A**) Midpoint-rooted phylogenetic tree constructed using amino acid sequences of the polyprotein (1000 bootstrap replicates; nodes with >70% bootstrap support are marked). Scale bar represents the number of amino acid substitutions per site. Discovered viruses are marked in red. (**B**) Scheme of Medvezhye pound Haematopota permuto-like virus genome. RdRp-encoding ORF is marked in green. All other ORFs are marked in purple.

**Figure 9 viruses-15-02368-f009:**
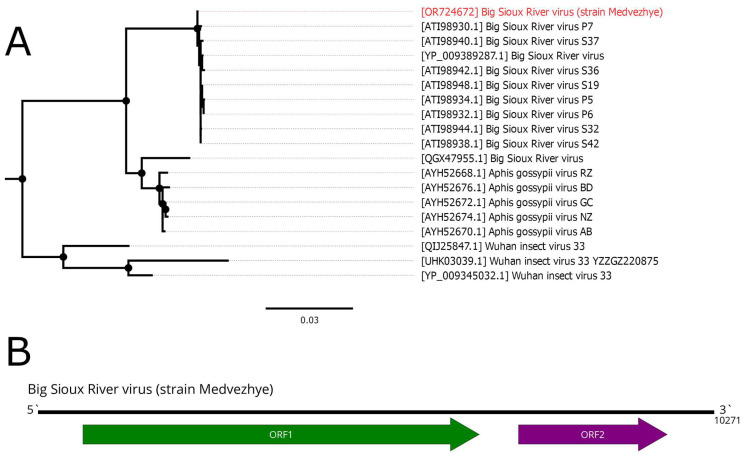
Genomic structure and phylogenetic relationships of Big Sioux River virus. (**A**) Midpoint-rooted phylogenetic tree constructed using amino acid sequences of the ORF1 (1000 bootstrap replicates; nodes with >70% bootstrap support are marked). Scale bar represents the number of amino acid substitutions per site. Discovered viruses are marked in red. (**B**) Scheme of Big Sioux River virus genome. RdRp-encoding ORF is marked in green. All other ORFs are marked in purple.

**Figure 10 viruses-15-02368-f010:**
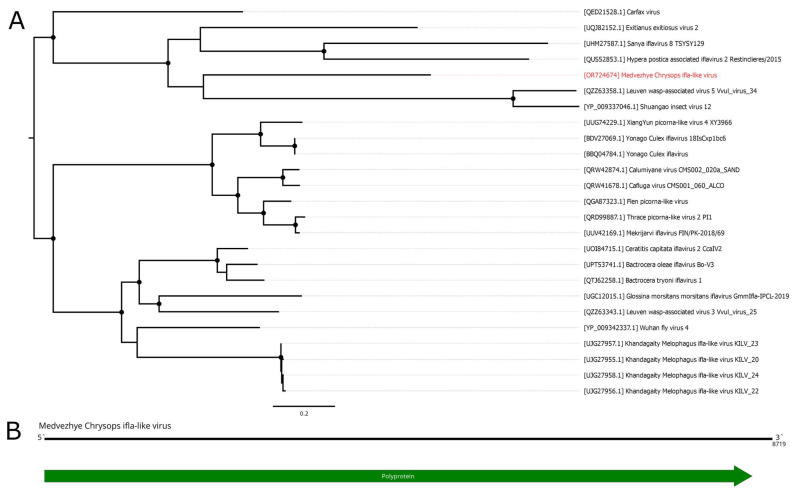
Genomic structure and phylogenetic relationships of Medvezhye Chrysops ifla-like virus. (**A**) Midpoint-rooted phylogenetic tree constructed using amino acid sequences of the polyprotein (1000 bootstrap replicates; nodes with >70% bootstrap support are marked). Scale bar represents the number of amino acid substitutions per site. Discovered viruses are marked in red. (**B**) Scheme of Medvezhye Chrysops ifla-like virus genome. RdRp-encoding ORF is marked in green.

**Table 1 viruses-15-02368-t001:** Collection and pooling of tabanids.

Pool Number	Region	Species in the Pool	Specimen Number	Location	Date
1	Primorye Territory	*Hybomitra brevis*	10	44.88123° 131.94521° 43.64188° 131.99379° 43.36378° 131.72099°	5–7 June 2021
2	Primorye Territory	*Hybomitra nigricornis*	4	43.88123° 131.94521°	7 June 2021
3	Primorye Territory	*Hybomitra stigmoptera*	10	43.64188° 131.99379°	8 June 2021
4		*Hybomitra lundbecki sibirensis*	2	44.88123° 131.94521°	5 June 2021
	Primorye Territory	*Hybomitra nitidifrons*	1	43.64188° 131.99379°	6 June 2021
		*Hybomitra astur*	2	43.88123° 131.94521°	6 June 2021
		*Hybomitra tarandina*	2	43.88123° 131.94521°	7 June 2021
5		*Hybomitra tarandina*	1	43.64188° 131.99379°	8 June 2021
	Primorye Territory	*Hybomitra brevis*	1		8 June 2021
		*Hybomitra lapponica*	1		8 June 2021
6	Ryazan Region	*Haematopota pluvialis*	10	54.306009° 40.035676°	5 July–13 August 2021
7	Ryazan Region	*Chrysops relictus*	10	54.306009° 40.035676°	5 July–13 August 2021
8	Ryazan Region	*Chrysops pictus*	4	54.306009° 40.035676°	5 July–13 August 2021
		*Chrysops caecutiens*	2	54.306009° 40.035676°	
9	Ryazan Region	*Tabanus autumnalis*	3	54.306009° 40.035676°	5 July–13 August 2021
		*Tabanus bromius*	2	54.306009° 40.035676°	
10	Ryazan Region	*Hybomitra schineri*	7	54.306009° 40.035676°	5 July–13 August 2021
		*Hybomitra muehlfeldi*	3	54.306009° 40.035676°	

**Table 2 viruses-15-02368-t002:** List of viruses detected in this study.

Pool Number	Virus Name	Assembly	Abundance *	GenBank
Pool 1	Xanka Hybomitra Negev-like virus	complete coding	0.01%	OR724662
	Razdolnyj Hybomitra Phasma-like virus	fragments	80 reads	OR724689
	28S rRNA ***	-	23.73%	-
Pool 2	Kamenushka Hybomitra Narna-like virus	complete coding	287 reads	OR724665
	Volxa Hybomitra Toti-like virus	complete coding	0.01%	OR724670
	Komarovka Hybomitra Solemo-like virus	partial **	394 reads	OR724676, OR724677
	Barsukovka Hybomitra Ifla-like virus	fragments	269 reads	OR724690
	28S rRNA	-	26.36%	-
Pool 3	Big rock Hybomitra Ifla-like virus	fragments	8 reads	OR724691
	28S rRNA	-	22.00%	-
Pool 4	28S rRNA	-	22.60%	-
Pool 5	28S rRNA	-	48.00%	-
Pool 6	Medvezhye Haematopota Flavi-like virus	complete coding	0.29%	OR724673
	Melisia Haematopota Negev-like virus	complete coding	0.23%	OR724663
	Medvezhye Haematopota Xinmo-like virus	complete coding	0.18%	OR724669
	Medvezhye pound Haematopota Permuto-like virus	complete coding	0.03%	OR724675
	Polka Haematopota Nora-like virus	fragments	14 reads	OR724692
	Polka Haematopota Xinmo-like virus	fragments	23 reads	OR724693
	28S rRNA	-	39.35%	-
Pool 7	Medvezhye Chrysops Negev-like virus	complete coding	1.08%	OR724664
	Big Soux River virus (Medvezhye strain)	complete coding	0.48%	OR724672
	Medvezhye Chrysops Solemo-like virus	complete coding	0.13%	OR724678, OR724679
	Melisia Chrysops Solemo-like virus	complete coding	0.28%	OR724680, OR724681
	Polka Chrysops Solemo-like virus	complete coding	0.05%	OR724682, OR724683
	Medvezhye Chrysops Narna-like virus	complete coding	0.61%	OR724666
	Medvezhye Chrysops Narna-like virus 2	complete coding	0.23%	OR724667
	Medvezhye Chrysops Solinvi-like virus	fragments	131 reads	OR724694
	Medvezhye Chrysops Rhabdo-like virus	fragments	0.01%	OR724695
	Melisia Chrysops Rhabdo-like virus	fragments	0.01%	OR724696
	28S rRNA	-	75.12%	-
Pool 8	Medvezhye Chrysops Ifla-like virus	partial	0.02%	OR724674
	Medvezhye Chrysops Toti-like virus	complete coding	0.03%	OR724671
	Istie Chrysops Solemo-like virus	partial	0.01%	OR724684, OR724685
	Medvezhye Chrysops Xinmo-like virus	partial	0.01%	OR724668
	Medvezhye Chrysops Phasma-like virus	fragments	0.02%	OR724686, OR724687, OR724688
	28S rRNA	-	64.74%	-
Pool 9	Medvezhye Tabanus Phasma-like virus	fragments	87 reads	OR724697
	Medvezhye Tabanus Toti-like virus	fragments	42 reads	OR724698
	Medvezhye Tabanus Chu-like virus	fragments	80 reads	OR724699
	28S rRNA	-	21.07%	-
Pool 10	28S rRNA	-	24.63%	-

* Percentage of total reads or amount of reads (if percentage is less than 0.01%). ** Gaps are estimated to be less than 10% of the coding sequence. *** Abundance of the largest contig containing tabanid 28S rRNA sequence (positive control).

**Table 3 viruses-15-02368-t003:** Detection of the viruses in the PEK and C6/36 cell cultures (after 3 passages) and HAE/CTVM8 (after 3 weeks of persistence).

Pool	Virus	Cell Cultures		
		HAE/CTVM8	C6/36	PEK
1	Xanka Hybomitra negev-like virus	+	+	+
2	Kamenushka Hybomitra narna-like virus	+	−	+
	Volxa Hybomitra toti-like virus	+	+	+
	Komarovka Hybomitra solemo-like virus	+	−	−
	Barsukovka Hybomitra ifla-like virus	+	−	−
6	Medvezhye pound Haematopota permuto-like virus	+	+	+
	Polka Haematopota nora-like virus	+	−	−
7	Medvezhye Chrysops negev-like virus	+	+	−
	Polka Chrysops solemo-like virus	+	+	−
	Medvezhye Chrysops solemo-like virus	+	−	−
	Big Sioux River virus (Medvezhye strain)	+	+	+
	Melisia Chrysops solemo-like virus	+	+	+
	Medvezhye Chrysops narna-like virus 2	+	+	+
	Medvezhye Chrysops rhabdo-like virus	−	+	−
8	Medvezhye Chrysops ifla-like virus	−	−	+
	Istie Chrysops solemo-like virus	+	+	−
9	Medvezhye tabanus Toti-like virus	+	+	+

“+”—virus was detected in cell culture supernatant using PCR. “−“—virus was not detected in cell culture supernatant using PCR.

**Table 4 viruses-15-02368-t004:** Detection of viruses in the PEK cell culture after 3 passages using high-throughput sequencing.

Pool	Reads Total	Virus	Virus Reads Amount
2	8627579	**Kamenushka Hybomitra Narna-like virus ***	45
		**Volxa Hybomitra Toti-like virus**	375
		Komarovka Hybomitra Solemo-like virus	66
7	8908134	Medvezhye Chrysops Negev-like virus	117
		Medvezhye Chrysops Solemo-like virus	9
		**Big Sioux River virus (Medvezhe strain)**	164
		**Melisia Chrysops Solemo-like virus**	25
9	9808163	**Medvezhye Tabanus Toti-like virus**	4

* viruses detected in PEK cell line using both PCR (Table 3) and high-throughput sequencing are marked in bold.

## Data Availability

Raw high-throughput sequencing data obtained during this study are available in the SRA database (BioProject accession number PRJNA1026651). Obtained virus sequences were deposited in the GenBank database (accession numbers OR724662—OR724699) and provided as an additional Appendix A.

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
