# Peer review of "Viromes of Tabanids from Russia"

_viruses, 2023, doi:10.3390/v15122368_

Round 1

Reviewer 1 Report

Comments and Suggestions for Authors

Viromes of several important groups of haematophagous insects are poorly characterized despite their importance and potential to harbor viruses capable to infect vertebrates. In this study high-throughput sequencing was used to characterize viromes of several species of several genera of  the family Tabanidae in order Diptera collected in two regions of Russia located over 6000 km apart, in Eastern European and Pacific region.  Bioinformatic analysis identified sequences (nearly full-length and partial) of 30 novel viruses. Infectivity of these novel virus was tested in mammalian, mosquito and tick cell cultures and persistence of some of these novel viruses in cell cultures  was demonstrated. 

The study presents novel RNA virus sequences and will be of interest to a wider virology community and contribute to our understanding of virus complexes in insects. This definitely warrants publication in Viruses. I have only few points which should be addressed:

Although novel sequences were submitted to GenBank (L, 784: " accession numbers OR724662 - OR724699 ",) these sequences are not accessible in GenBank yet. Considering that the novel virus sequence data is the most important output of this study, all these nucelotide sequences should be included as a Supplement text to be immediately available for an independent analysis. 

Abstract. L.17-18

Specify in which regions the insects were collected. The samples were gathered in Easter Europe and Far East, these sites are located approximately 6 thousand kilometers apart in different climate areas, and importantly, different species were tested for these locations. 

L. 24. (Nevertheless, four viruses were detected after six passages.)

Specify in which cell lines and which viruses.  

L203-213 and Figure 1. 

I suggest, in addition to Fig. 1,  to include a main text Table which will to give numbers of reads  which for each of these virus groups. This table should also include an "internal control", numbers of the reads derived from an abundant host mRNA (for example beta-actin mRNA) to demonstrate rate that the absence of viral reads in some libraries was not due to a technical sequencing failure. 

L.210 "...Negevirus group, Narnaviridae, ..." -> "...Negevirus group, families Narnaviridae, ..."

L. 210-212, Table 1. Make sure that the lists of virus groups mentioned in the lines 210-212 and in Fig. 1 are matching. Currently, Phasmaviridae is in the text but not mentioned in  Fig.1,  and Orthophasma-like and Nora-like re in Fig. 1 but not in the text. 

Figures 2, 6 , 7, 8, 

Specify sequence of which proteins was used for phylogenetic analyses. 

Author Response

Reviewer 1

Comment:

Viromes of several important groups of haematophagous insects are poorly characterized despite their importance and potential to harbor viruses capable to infect vertebrates. In this study high-throughput sequencing was used to characterize viromes of several species of several genera of  the family Tabanidae in order Diptera collected in two regions of Russia located over 6000 km apart, in Eastern European and Pacific region.  Bioinformatic analysis identified sequences (nearly full-length and partial) of 30 novel viruses. Infectivity of these novel virus was tested in mammalian, mosquito and tick cell cultures and persistence of some of these novel viruses in cell cultures  was demonstrated. 

The study presents novel RNA virus sequences and will be of interest to a wider virology community and contribute to our understanding of virus complexes in insects. This definitely warrants publication in Viruses. I have only few points which should be addressed:

Answer:

We want to thank the reviewer for the kind comments.

Comment:

Although novel sequences were submitted to GenBank (L, 784: " accession numbers OR724662 - OR724699 ",) these sequences are not accessible in GenBank yet. Considering that the novel virus sequence data is the most important output of this study, all these nucelotide sequences should be included as a Supplement text to be immediately available for an independent analysis. 

Answer:

Thank you for the suggestion. We added additional Supplementary file, containing all deposited sequences.

Comment:

Abstract. L.17-18

Specify in which regions the insects were collected. The samples were gathered in Easter Europe and Far East, these sites are located approximately 6 thousand kilometers apart in different climate areas, and importantly, different species were tested for these locations. 

Answer:

We specified the regions in the abstract.

Comment:

  1. 24. (Nevertheless, four viruses were detected after six passages.)

Specify in which cell lines and which viruses.  

Answer:

Thank you for the comment. The line was deleted.

This line in was left in the abstract by mistake since the earlier version of the manuscript. Sanger sequencing did not confirm the presence of the virus RNA after six passages, as was stated in the Results section (see lines 726-728).

We are very sorry for the confusion.

Comment:

L203-213 and Figure 1. 

I suggest, in addition to Fig. 1, to include a main text Table which will to give numbers of reads  which for each of these virus groups. This table should also include an "internal control", numbers of the reads derived from an abundant host mRNA (for example beta-actin mRNA) to demonstrate rate that the absence of viral reads in some libraries was not due to a technical sequencing failure. 

Answer: We added the Table in the main text, describing viruses discovered and their abundance, sequences of the 28S RNA was used as a positive control.

Comment:

L.210 "...Negevirus group, Narnaviridae, ..." -> "...Negevirus group, families Narnaviridae, ..."

Answer:

The text was corrected.

Comment:

  1. 210-212, Table 1. Make sure that the lists of virus groups mentioned in the lines 210-212 and in Fig. 1 are matching. Currently, Phasmaviridae is in the text but not mentioned in  Fig.1,  and Orthophasma-like and Nora-like re in Fig. 1 but not in the text. 

 Answer:

Family Noraviridae was added to the list in the text, ‘Orthophasma-like’ was changed into ‘Phasma-like’ in the Figure 1.

Comment:

Figures 2, 6, 7, 8, Specify sequence of which proteins was used for phylogenetic analyses.

Answer:

Thank you for the comment. We adjusted the descriptions in the Figures 2, 6, 7 by specifying protein used for the analysis. Figure 8 already has correct description, since the whole polyprotein (see Figure 8B) was used for phylogenetic analysis. 

It should be noted, that description of all figures were additionally adjusted according to the editors comments.

Reviewer 2 Report

Comments and Suggestions for Authors

The manuscript entitled: Virome of the tabanids from Russia by Litov et al., represents valuable research on the RNA viromes of Hybomitra, Tabanus, Chrysops, and Haematopota genera collected in the Primorye Territory and Ryazan Region, using high-throughput sequencing. In addition, the discovered viruses were tested on their ability to replicate in the mammalian porcine embryo kidney (PEK), tick HAE/CTVM8, and mosquito C6/36 cell lines. I would recommend '' Tabanids'' instead of ''tabanids'' in the title.

The methodology and results are well executed and presented.  The discussion is well-structured and includes newly identified viruses, where 14 of them had complete coding sequences assembled, and can be accepted for further taxonomic classification.

Author Response

Reviewer 2

Comment:

The manuscript entitled: Virome of the tabanids from Russia by Litov et al., represents valuable research on the RNA viromes of HybomitraTabanusChrysops, and Haematopota genera collected in the Primorye Territory and Ryazan Region, using high-throughput sequencing. In addition, the discovered viruses were tested on their ability to replicate in the mammalian porcine embryo kidney (PEK), tick HAE/CTVM8, and mosquito C6/36 cell lines. I would recommend '' Tabanids'' instead of ''tabanids'' in the title.

The methodology and results are well executed and presented.  The discussion is well-structured and includes newly identified viruses, where 14 of them had complete coding sequences assembled, and can be accepted for further taxonomic classification.

Answer:

We want to thank the reviewer for his kind comments. The title was adjusted according to the recommendations.